# Design of Destruction Protection and Sustainability Low-Dropout Regulator Using an Electrostatic Discharge Protection Circuit

Sang-Wook Kwon , Seung-Gu Jeong, Jeong-Min Lee and Yong-Seo Koo *

Department of Electronics and Electrical Engineering, Dankook University, Yongin-si 16890, Republic of Korea; kso804@naver.com (S.-W.K.); aine6345@naver.com (S.-G.J.); aint0ver5@naver.com (J.-M.L.)
* Correspondence: yskoo@dankook.ac.kr

**Abstract:** In terms of sustainable power semiconductors, the embedding of an electrostatic discharge (ESD) protection circuit in an integrated circuit (IC) is an important aspect. In order for the semiconductor circuit to operate continuously or stably, a sufficient protection circuit against external surges must be configured. The purpose of this thesis is not only to effectively operate the low-dropout (LDO) regulator according to the load current, but to also secure high reliability against ESD situations by embedding an ESD protection circuit at the IC level. Moreover, the existence and nonexistence of an ESD protection circuit at the IC level is directly related to reliability. The proposed LDO regulator has high reliability against ESD situations using an embedded silicon controlled rectifier (SCR)-based ESD protection circuit in the I/O clamp and power clamp. The results revealed that the LDO regulator can not only effectively control the output voltage according to the load current, but it can also stably maintain the output voltage against the ESD surge. Moreover, the proposed LDO regulator with an embedded ESD protection circuit implemented in a 0.13 μm BCD process maintained an undershoot voltage of 21 mV and overshoot voltage of 19 mV for a load current of 300 mA.

**Keywords:** LDO regulator with ESD protection circuit; high-reliability circuit; IC protection; sustainable ESD protection

## 1. Introduction

In terms of sustainable power semiconductors, the embedding of an ESD protection circuit within an IC is crucial. To ensure continuous and reliable operation of the semiconductor circuit, it is essential to construct a reliable protection circuit against external surges. ESD events occur frequently in everyday situations. In general, people are not significantly affected by ESD events. However, in terms of power semiconductors, if the IC is exposed to ESD conditions without protection, it will be destroyed. Circuit destruction due to ESD conditions disables normal circuit operation, resulting in total system destruction. Figure 1 shows the reliability depending on the voltage level of the LDO regulator application with or without an ESD protection circuit. LDO regulator applications with an ESD protection circuit can secure high reliability and sustainability by safely protecting the IC when an ESD phenomenon occurs. However, LDO regulator applications without an ESD protection circuit cannot secure high reliability and sustainability because they are unable to safely protect the IC in the event of an ESD phenomenon. Therefore, if a mobile or wearable device is used without configuring an ESD protection circuit, the normal operation of the device cannot be guaranteed due to frequent circuit destruction. It can be confirmed that the presence or absence of an ESD protection circuit, as a result of an ESD event, significantly affects the reliability and sustainability of the circuit. As a result, in terms of sustainable power semiconductors, the construction of an ESD protection circuit is essential. Wearable and mobile application markets are growing to new heights every year. Systems using various applications for convenient daily life require high performance that can be

supported for a long time with limited battery capacity. Due to the varying voltage and current conditions used in these applications, a load condition is created that causes the power supply to fluctuate instantaneously. Because of the huge required quiescent current, mobile applications require voltage stability over the load current. LDO regulators used in low-voltage applications are among the most cost-effective control components for systems requiring a variable voltage and load current [1]. If an unstable output voltage is provided by the LDO regulator as the load current changes, the voltage required by the system constantly changes, which has a fatal effect on the IC operation. Therefore, as shown in Figure 2, the LDO regulator must be configured to operate reliably regardless of changes in the load current. As the use of electronic devices has become increasingly common in everyday life, ESD is becoming a more significant issue. While momentary ESD surge has harmless effects on the human body, the IC can be damaged or destroyed, causing permanent damage due to the high voltages and currents of the ESD surge. Therefore, the proposed ESD protection circuit is designed to protect the LDO regulator from the harmful effects of the ESD. Such an ESD protection circuit must be embedded to ensure the reliability and lifespan of the IC, such as those used in LDO regulators. Also, the LDO regulators presented in the previous papers do not have an ESD protection circuit [2–4]. If each circuit is individually designed and operated, this is not a problem. However, dozens of LDO regulators are designed in integrated circuits of power semiconductors. All electronic devices used by actual customers are exposed to ESD situations. ESD situations are an extremely common occurrence. Therefore, it is essential to consider the ESD situation. Even if the ESD condition is intentionally given, if the LDO regulator operates normally, the ESD condition can be effectively protected with a smaller area. In terms of sustainable power semiconductor system design, an LDO regulator with an embedded ESD protection circuit is an essential element that must be developed for next-generation power semiconductors. The reason is that exposure to ESD situations frequently occurs due to miniaturization of the process when looking at not only power semiconductors but also semiconductors as a whole. Therefore, in this paper, an ESD protection circuit is embedded in the proposed LDO regulator to verify the high reliability and sustainability of power semiconductors in ESD situations.

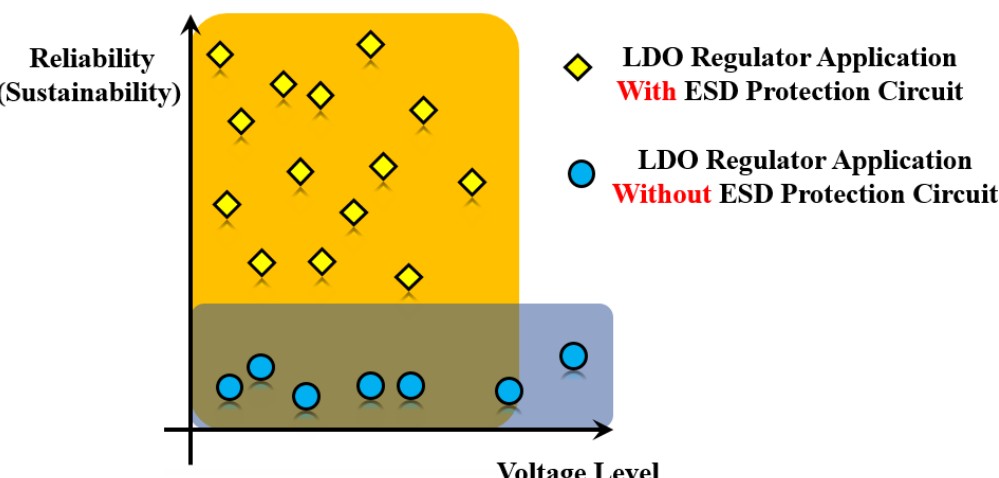

**Figure 1.** Difference in reliability and sustainability of LDO Regulators with and without ESD protection circuit.

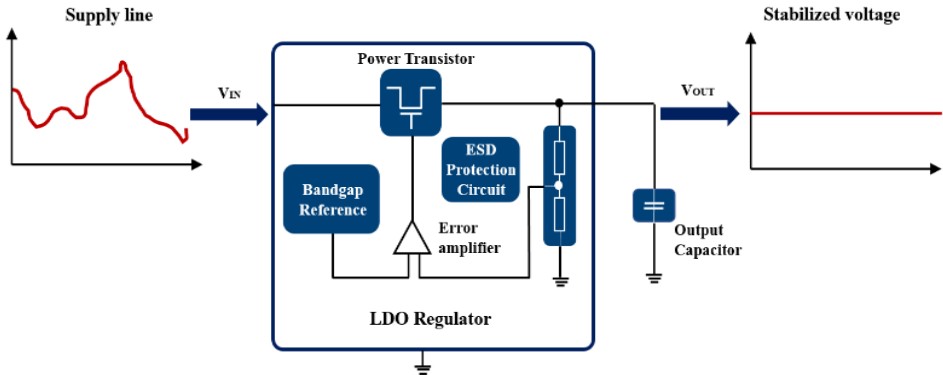

**Figure 2.** Voltage regulation using LDO regulator.

## 2. Proposed LDO Regulator with Transient Current-Sensing Structure

As shown in Figure 3, the proposed LDO regulator consists of a transient current-sensing structure and an SCR-based ESD protection circuit. The transient current-sensing structure was designed to detect load current variations and form additional current paths. Additionally, it is important to embed an ESD protection circuit between the VDD-VSS and VOUT-VSS of the LDO regulator. When an ESD surge is applied to the VDD node, the breakdown voltage of the internal components of the LDO regulator can be exceeded, damaging the LDO regulator's transistors and causing them to malfunction. Therefore, the embedded ESD protection circuit between VDD and VSS and the stable discharge of current to the ESD surge protects the LDO regulator from damage and ensures stable operation. In addition, the output node of the LDO regulator is connected to the system load or to other IC systems. An ESD surge applied to the output node can exceed the allowable voltage levels of the IC to which the LDO regulator is connected, damaging or completely destroying the component. The embedded ESD protection circuit between VOUT and VSS isolates the ESD current from the connected load and IC to prevent damage and ensure stable operation. Therefore, an ESD protection circuit is embedded in the VDD-VSS and VOUT-VSS to quickly respond to ESD events and mitigate their effects. Consequently, the proposed LDO regulator is designed to maintain voltage regulation stability and ensure that the connected IC receives a stable power supply.

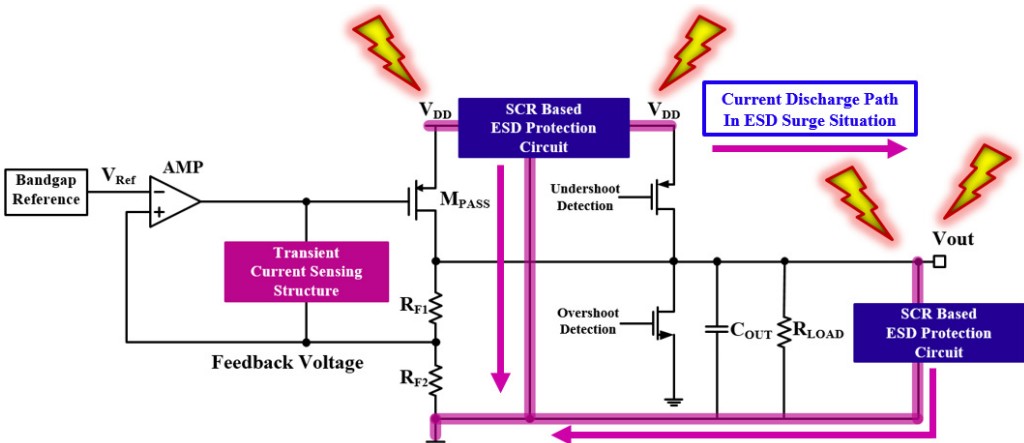

**Figure 3.** The LDO regulator using transient current-sensing structure and current path in ESD situation.

Figure 4 shows the operation of the transient current-sensing structure as the load current rapidly increases. The transient current-sensing structure senses the feedback voltage because it is sensitive to changes in the load current [5]. The rapid increase in the load current means that the output voltage of the LDO regulator has decreased. As the load current rapidly increases, the feedback voltage is lowered, and the transient current-sensing

structure supplies the discharge current to the pass transistor of the LDO regulator while supplying the supply current to the output node of the LDO regulator [6]. Figure 5 shows the transient current-sensing structure when the load current rapidly decreases.

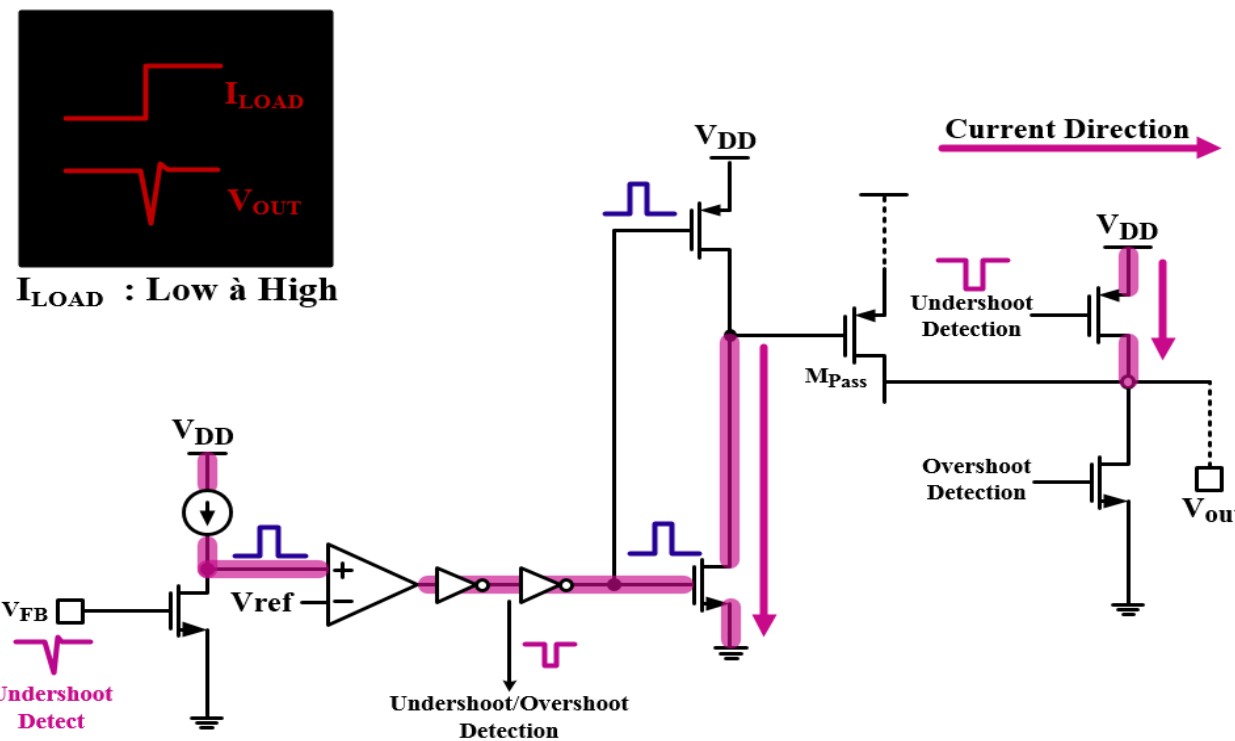

**Figure 4.** Transient current sensing according to undershoot situation.

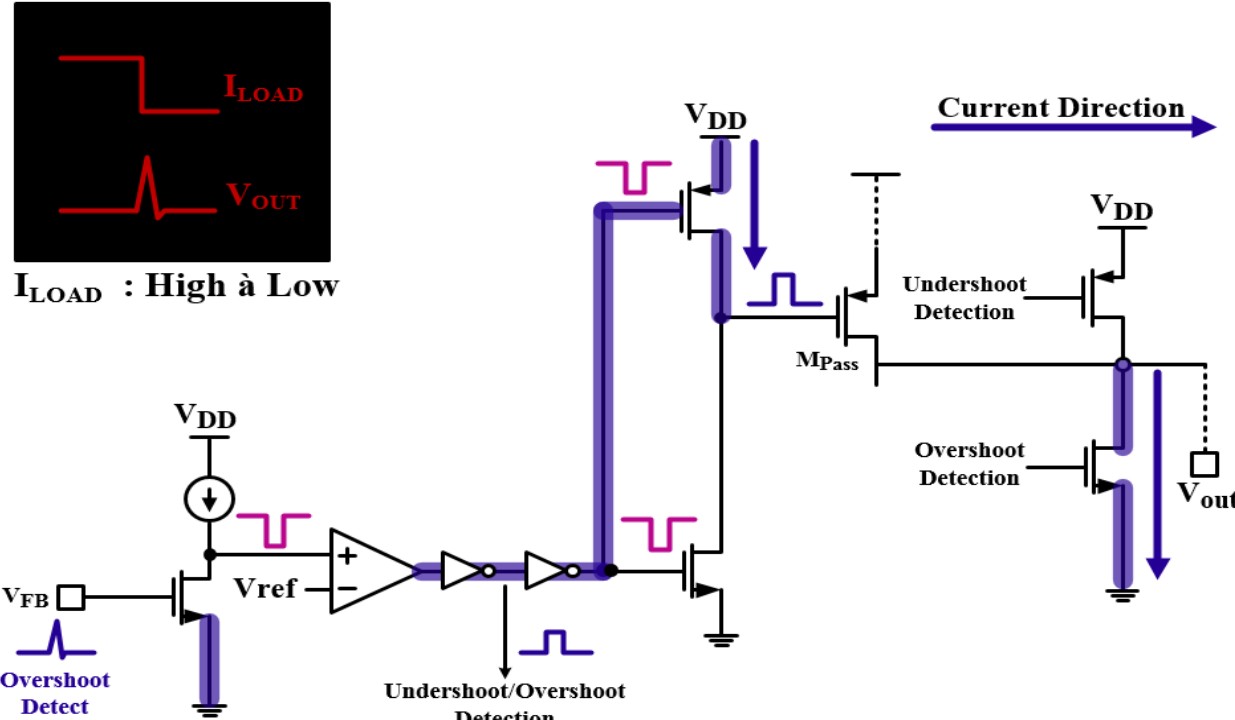

**Figure 5.** Transient current sensing according to overshoot situation.

The rapid decrease in the load current means that the output voltage of the LDO regulator has increased [7]. When the load current decreases, the feedback voltage increases, supplying a current to the LDO regulator's pass transistor through the transient sensing structure while forming a discharge current at the output node of the LDO regulator. Typically, the pass transistor of the LDO regulator is designed to be as sizable as possible. This indicates that the pass transistor has the most superior current driving capability among the LDO regulators [8]. Hence, the proposed LDO regulator was designed to effectively control the load current by discharging and supplying additional current to the gate node of the pass transistor [9]. Simultaneously, the proposed LDO regulator was designed to supply and discharge additional current according to the load current fluctuations at the output node. The output node of the LDO regulator is a node that reacts sensitively to the load current. Therefore, it is designed to effectively control the output voltage by supplying and discharging additional current to the output node. Consequently, the proposed LDO regulator can efficiently control the output voltage by forming a path that can supply and discharge current to a node that affects the output voltage [10].

## 3. LDO Regulator Circuit Configuration with Embedded ESD Protection Circuit for Sustainable Power Semiconductor Configuration

In terms of sustainable power system design, LDO regulators must solve sustainability more than necessary. If the LDO regulator is designed not to guarantee sustainability, the stable operation of the IC circuit will be impossible. The biggest factor affecting the sustainability of LDO regulators is circuit destruction due to an ESD phenomenon. If the internal circuit of the LDO regulator is destroyed by the ESD phenomenon, it will not only fail to ensure the normal operation of the circuit where the ESD phenomenon has occurred, but it will also cause a fatal operating error in the entire system circuit, which can impair sustainability [11]. Sustainability in the circuit is related to the extension of the lifespan of the circuit, and lifespan is directly related to the reliability of the circuit. Therefore, in order to pursue the sustainability of the circuit, it is essential to secure the stability and reliability of the circuit by preventing device failure or serious operation interruption caused by ESD. Figure 6 shows embedded the ESD protection circuit configuration for the ESD event of the LDO regulator. When an ESD phenomenon occurs, the ESD protection circuit operates to prevent ESD current from flowing into the internal circuit through the I/O clamp and power clamp and discharges the ESD current to the VSS pin to maintain the internal circuit sustainably. The ESD protection circuit operates at high voltages and must be able to protect the internal circuit [12]. In addition, a current driving ability to discharge a large current in a small area is required, and it should be designed not to affect normal operation by being turned off during normal operation [13]. Figure 7 shows the embedded diode configuration for LDO regulator ESD events. A diode is a device commonly used as an ESD protection device. A diode is widely used as an ESD protection circuit because it is easy to implement and design because of its simple structure. However, diodes have some major drawbacks, the first being area. In general, when a diode is in the forward direction, it has a turn-on voltage of around 0.5 V to 1 V and discharges current. Therefore, ESD protection via a diode must be designed using a string diode structure using several diodes. As shown in Figure 7, the area of the diode is 23.24 μm × 23.24 μm. Assuming that the forward voltage of the diode is generally 0.7 V, at least eight diodes are needed to use it as an ESD protection circuit of 5 V or more. That is, it means that it occupies an area of 23.24 μm × 23.24 μm × 8 or larger. This is larger than the area of 64.4 μm × 57.4 μm of the ESD protection circuit shown in Figure 6. The second drawback is that the actual turn-on voltage of a PN diode is very sensitive to temperature. Generally, at room temperature, if the forward turn-on voltage of a diode is 0.7 V, in a high-temperature environment, the forward turn-on voltage of a diode is about 0.3 V or 0.4 V, which is significantly reduced compared to 0.7 V. This means that the characteristics of the ESD discharge can be easily changed, and in other words, it is an important issue directly relating to the sustainability of the circuit. Finally, one of the biggest problems with a diode is that it has poor robustness compared to an ESD protection

circuit of the same size. If the size of the HBM applied to the diode increases, the diode itself may be destroyed. In order to withstand a high level of HBM, there is a fatal disadvantage in that the number of diodes must be increased, or the area of the diode must be sufficiently large. An LDO regulator with an embedded ESD protection circuit has several advantages. First of all, the low-dropout voltage characteristic of LDO regulators minimizes power losses, improving the overall energy efficiency of electronic systems and contributing to energy sustainability. Second, embedded ESD protection protects devices from potential damage from electrostatic discharge, improving the longevity and reliability of devices critical to achieving sustainable development. Therefore, by embedding the proposed ESD protection circuit in the LDO regulator, the ESD current can be safely discharged to ensure stable operation of the internal circuit against ESD, and the overall design area can be reduced, making it suitable for space-limited applications. This miniaturization is consistent with pursuing sustainability by promoting efficient use of resources [14].

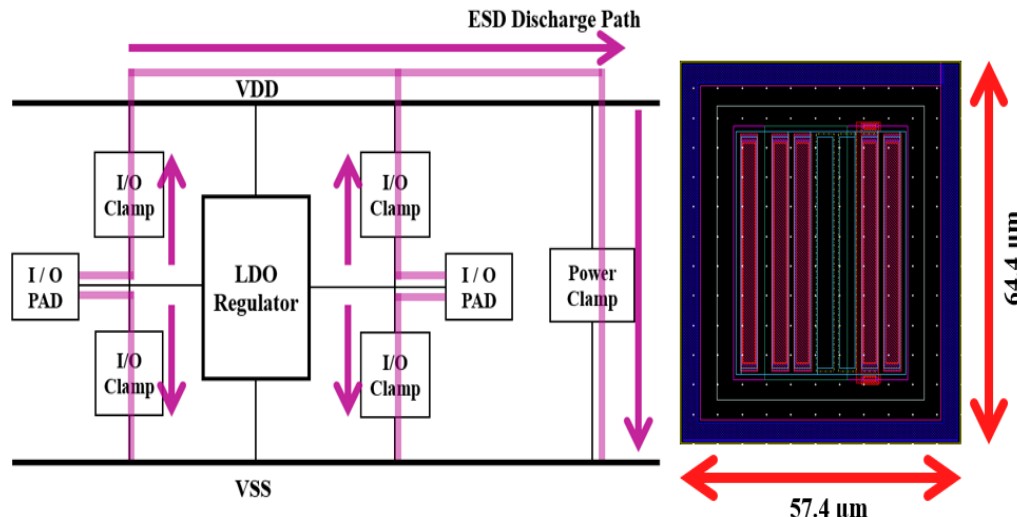

**Figure 6.** Embedded ESD protection circuit configuration for ESD event of LDO regulator.

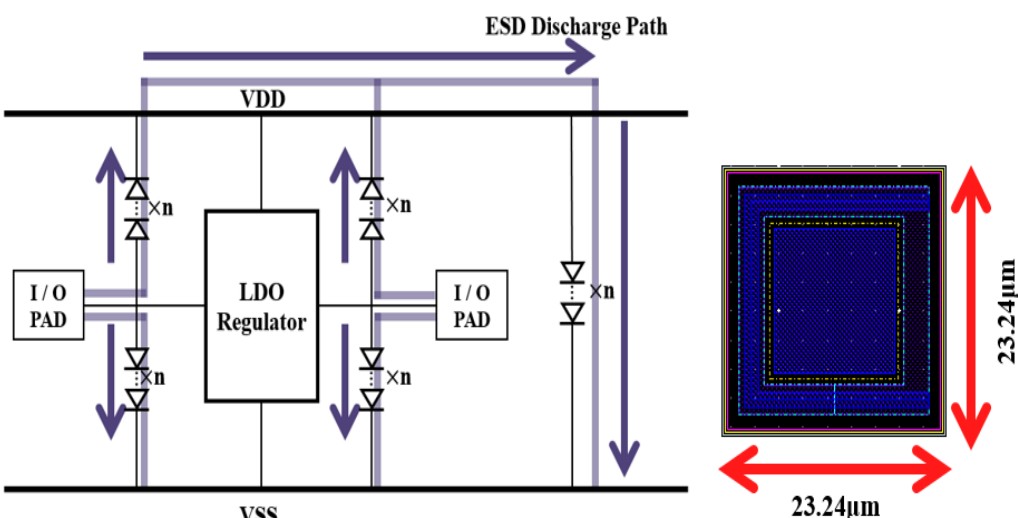

**Figure 7.** Embedded diode configuration for LDO Regulator ESD events.

Figure 8 shows the cross section and layout of the proposed ESD protection circuit. As a structural feature of the conventional low-Ron silicon-controlled rectifier (LRSCR), which is the base of the proposed ESD protection circuit structure, by inserting the grounded gate N-MOS (GGNMOS) structure into the conventional SCR structure, an avalanche breakdown happens in the N+ bridge and P-Well regions to lower the trigger voltage [15,16]. By

adding P-well and P+ implant regions to the anode region, an additional parasitic PNP bipolar transistor (Qpnp2) is operated, rather than the conventional SCR [17]. Therefore, by providing a parallel discharge path for the ESD current, the on-resistance characteristic is reduced, and the robustness characteristic is improved. The ESD protection circuit with a high holding voltage is proposed as the current gain is reduced by expanding the length of the base region of the parasitic PNP bipolar transistor by adding an N+ floating region to the conventional LRSCR structure having these structural features. The following is the operating mechanism of the proposed ESD protection circuit: The N+ bridge region, which has an elevated potential owing to the ESD surge from the anode region, causes an avalanche breakdown in the P-well region junction, creating an EHP. The generated holes move to the P+ cathode region and increase the potential of the P-well region. Therefore, the P-well and N+ cathode regions are in a forward-biased state, and the parasitic NPN bipolar transistor (Qnpn1) is turned on. The electrons generated at the same time as the holes move to the N+ anode region and reduce the potential of the N-well region. Accordingly, the N-well and P+ anode regions are forward biased, and the parasitic PNP bipolar transistor Qpnp1 is turned on. In addition, the P- and N-well regions added to the left become forward biased, and the additional parasitic PNP bipolar transistor (Qpnp2) is turned on [18]. Therefore, one parasitic NPN bipolar transistor and two parasitic PNP bipolar transistors that are turned on discharge the ESD current through the latch mode. Consequently, the proposed ESD protection circuit has a higher holding voltage by modifying the structure of the conventional LRSCR; therefore, it is embedded in the proposed LDO regulator to ensure high reliability and improved lifespan [19].

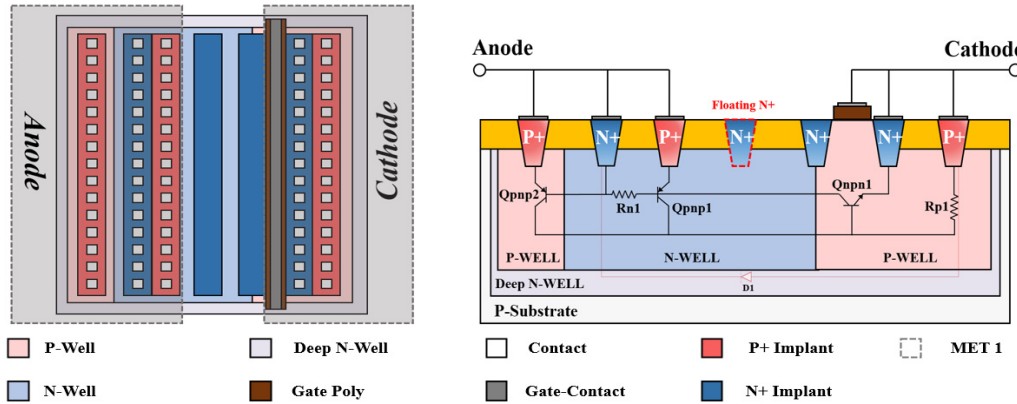

**Figure 8.** Cross-section of the embedded ESD protection circuit of the proposed LDO regulator.

## 4. Measurement Results

Figure 9 shows that the proposed LDO regulator is designed with a 0.13 μm Bipolar-CMOS-DMOS (BCD) process and has a chip size of 521 × 473 μm [20]. If a large peak voltage occurs, depending on the load current, it can affect the system connected to the output node of the LDO regulator. An LDO regulator with a transient current-sensing structure produces a small peak voltage according to the load current fluctuations [21]. The proposed LDO regulator may be affected by ESD events and causes damage and malfunction. Therefore, the proposed LDO regulator had an embedded ESD protection circuit, which was measured to verify its high reliability. An ESD zapping test to evaluate the high reliability of ESD was conducted via human body model (HBM) (4 KV, 6 KV, 8 KV) tests in 2 KV steps. To measure HBM, place a wafer on the probe station and apply the HBM pulse through a Noise Ken ESS 6008 simulator. The TLP System is a widely used measurement method to analyze the electrical characteristics and robustness characteristics of ESD protection devices. The principle of TLP measurement is that a square wave with a pulse width of 100 ns is applied to the device under test (DUT) with a step-by-step increase, and then, a method of measuring voltage and current using an oscilloscope is performed. Additionally, the measurement results of the LDO regulator were measured using an

oscilloscope, a DC power source, and a DC electronic load. The proposed LDO regulator can provide ESD immunity against ESD events by placing I/O clamp and power clamp on the VOUT-VSS and VDD-VSS lines, respectively [22]. In this paper, the ESD protection circuit-embedded case and the diode-embedded case were divided into measurements. Since the ESD protection circuit used in this paper is used as a 5 V class protection circuit, if a diode is used, more than eight diodes are required, which reduces the area efficiency [23]. In addition, the diode can withstand more than HBM 2 KV depending on the area, but for a more accurate comparison, a diode with a similar area to the embedded ESD protection circuit is embedded.

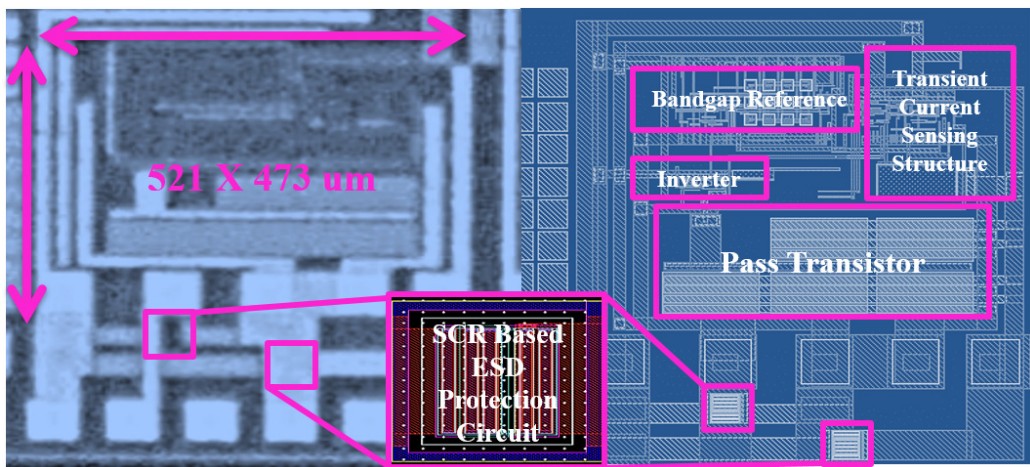

**Figure 9.** Chip layout of transient current-sensing structure LDO regulator including ESD protection circuit.

Figures 10 and 11 present the measurement results based on the embedding existence and nonexistence of the proposed ESD protection circuit to demonstrate the ESD protection effect of the proposed LDO regulator against ESD events. Figure 10 shows the transient response characteristics of the proposed LDO regulator without an embedded ESD protection circuit [24]. The LDO regulator does not operate normally, even under a small ESD condition of HBM = 2 KV. The proposed LDO regulator without an embedded ESD protection circuit failed to provide a normal operating voltage from the output voltage. In addition, it failed to provide a regulation operation as well as normal feedback operation, even when the load current changed. Figure 11 shows the transient response characteristics of the proposed LDO regulator embedded with the ESD protection circuit. The proposed LDO regulator was measured after applying the large ESD condition of HBM = 8 KV. The proposed LDO regulator maintained a small peak voltage of 21 mV under undershoot conditions and 19 mV under overshoot conditions, even with a large load current of 300 mA. Consequently, the proposed LDO regulator effectively controlled the output voltage owing to its transient current-sensing structure. In addition, the proposed LDO regulator embedded with an ESD protection circuit provided a stable output voltage under various ESD conditions. However, the measurement results of the LDO regulator without an embedded ESD protection circuit did not effectively discharge the ESD surge, causing the destruction of the internal system. Figure 12 shows the load regulation measurement results of the proposed LDO regulator using a transient current-sensing structure. In the ESD situation of HBM = 8 KV, the proposed LDO regulator embedded with an ESD protection circuit had a voltage change of 6.72 mV when the load current was continuously increased up to 300 mA. However, it can be confirmed that the LDO regulator without an embedded ESD protection circuit failed to provide a stable output voltage and did not perform a normal regulation operation.

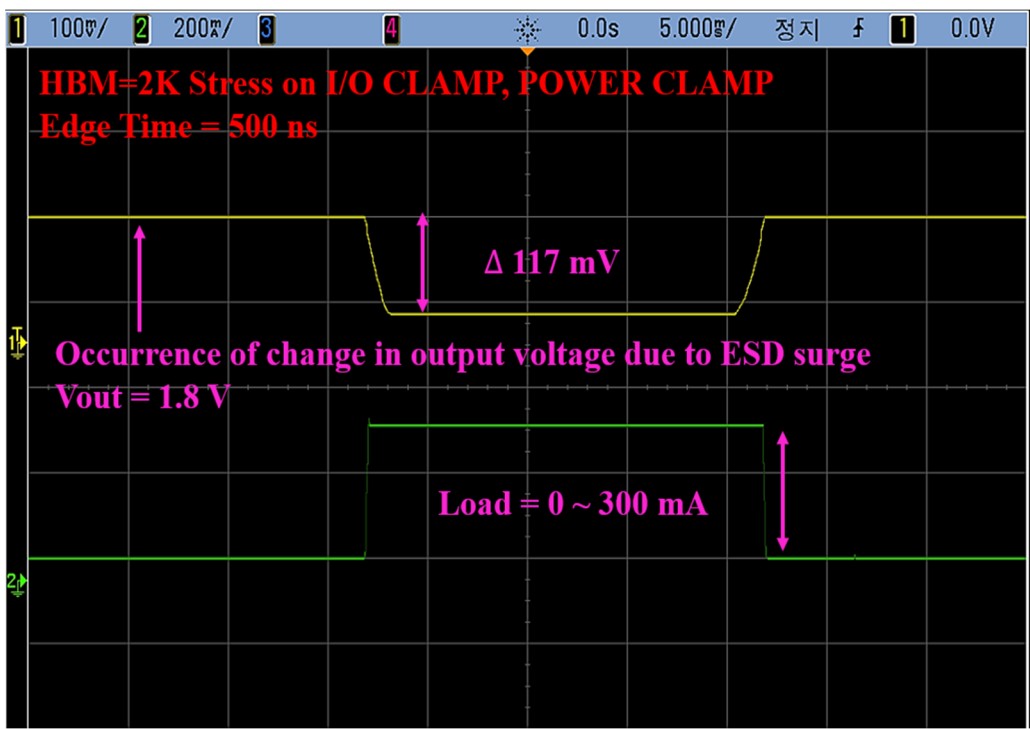

**Figure 10.** Transient response measurement result of transient current-sensing structure LDO regulator without proposed ESD protection circuit.

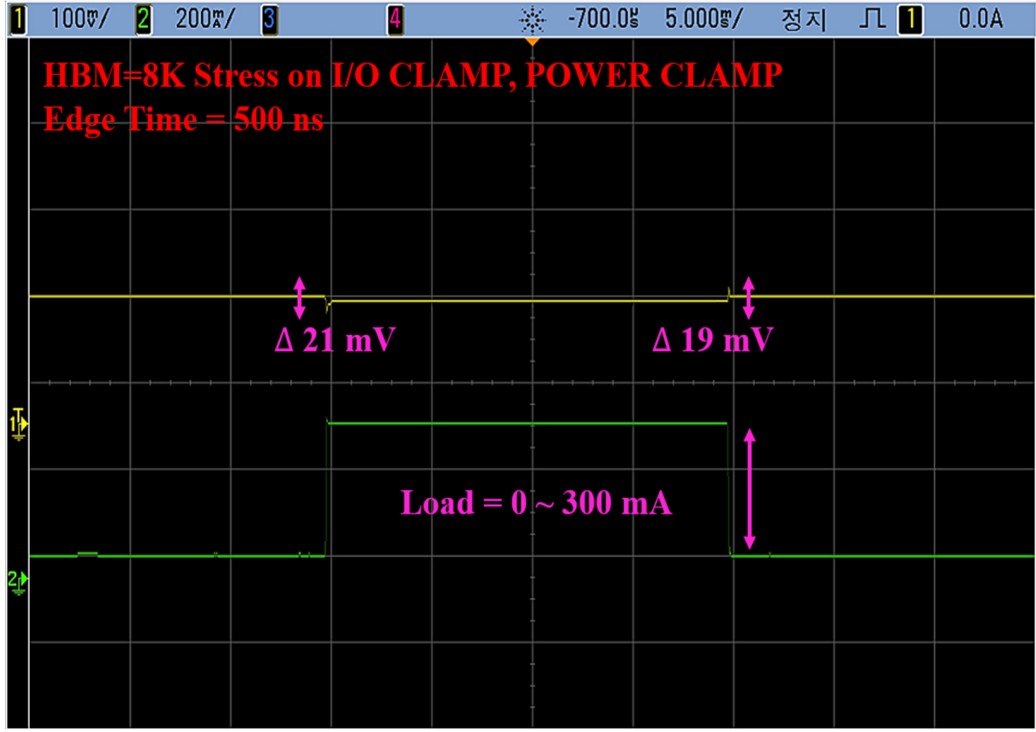

**Figure 11.** Transient response measurement result of transient current-sensing structure LDO regulator embedded with the proposed ESD protection circuit.

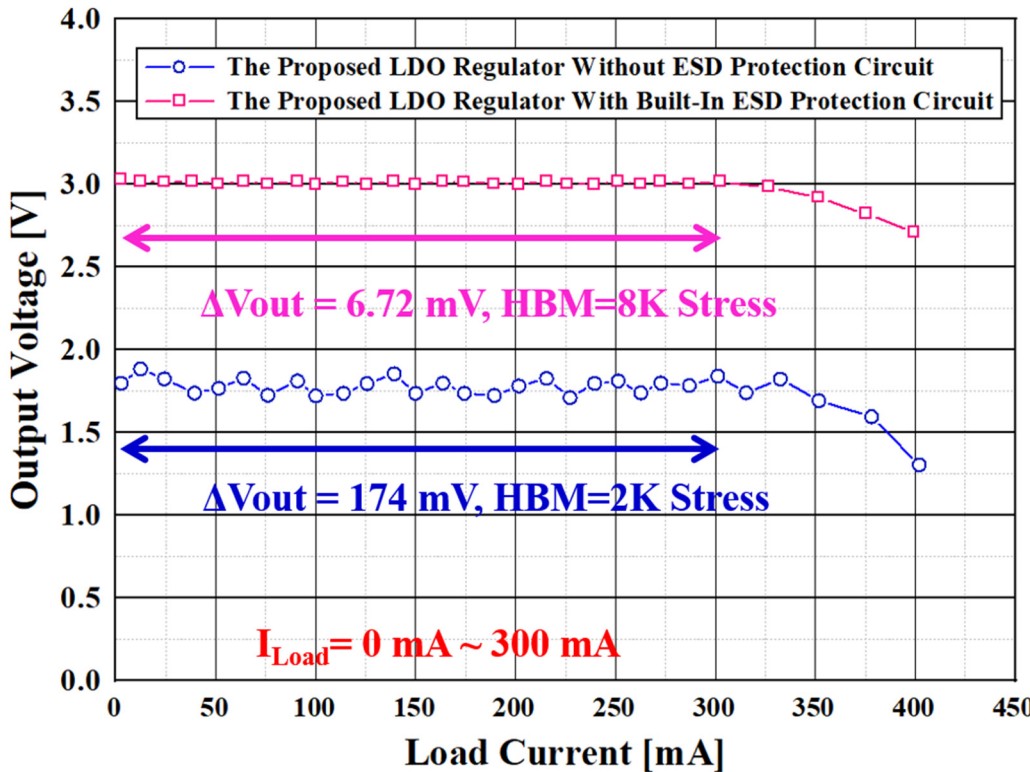

**Figure 12.** Load regulation measurement result of transient current-sensing structure LDO regulator (embedded with/without ESD protection circuit).

Figure 13 shows the line regulation measurement results for the proposed LDO regulator using a transient current-sensing structure. In general, since the output voltage of an LDO regulator is provided as the input voltage of other systems, it must provide a stable output voltage even when the input voltage changes [25,26]. It is confirmed that the proposed LDO regulator maintains a voltage change of 7.6 mV when the input voltage is constantly increased in the range of 3.3 V to 4.5 V. It was proven that the LDO regulator embedded with an ESD protection circuit stably controlled the output voltage under ESD conditions of HBM = 8 KV and various input voltage ranges. Figure 14 shows the proposed LDO regulator's temperature measurement results. In general, the output voltage of an LDO regulator should provide a stable output voltage despite changes in load current and input voltage, but it should not provide a sensitive output voltage even with temperature changes [27]. The reason is that in mobile and wearable devices, in general, as the amount of power increases, the temperature increases. If the output voltage reacts sensitively as the temperature of the device increases, the system will not be able to operate stably [28]. The proposed LDO regulator embedded with an ESD protection circuit confirmed a temperature change of 3.12 mV in the ESD situation of HBM = 8 KV. Conversely, the proposed LDO regulator without an embedded ESD protection circuit confirmed a temperature change of 39.2 mV in the ESD situation of HBM = 2 KV. Figure 15 shows the measurement results of the quiescent current of the proposed LDO regulator. The proposed LDO regulator with an embedded ESD protection circuit secured minimum quiescent current of 35 μA and maximum quiescent current of 38 μA in ESD situation with HBM = 8 KV. However, it can be confirmed that the LDO regulator without an ESD protection circuit does not guarantee circuit sustainability even in the ESD situation of HBM = 2 KV. Figure 16 shows the transmission line pulse (TLP) I-V measurement results of the proposed ESD protection circuit and the conventional LRSCR [29]. The ESD design window for the proposed LDO regulator and 5 V class application was formed between the operating voltage and 10% margin of the operating voltage and the core damage region (gate oxide breakdown = 12 V) [30]. According to the TLP measurement result, the holding

voltage of the conventional LRSCR is 2.9 V, which is lower than the maximum value of the operating voltage of the proposed LDO regulator of 4.5 V, which can cause serious problems such as latch-up by invading the operating range [31,32]. However, the proposed ESD protection circuit increased the holding voltage from 2.9 V to 5.8 V through a structural transformation of the conventional LRSCR; therefore, it is suitable for the 5 V class ESD design window and has a higher current driving capability through the excellent parallel discharge path [33,34]. In summary, since the holding voltage of the conventional LRSCR is lower than the proposed LDO regulator's maximum operating voltage of 4.5 V, a latch-up problem occurs [35]. So, even if the conventional LRSCR is embedded in the proposed LDO regulator, reliability cannot be guaranteed. Therefore, stable operation and high reliability were ensured by embedding the proposed ESD protection circuit in the proposed LDO regulator [36]. Table 1 presents a comprehensive comparison of the performance and efficiency between the proposed LDO regulator with an embedded ESD protection circuit and a conventional LDO regulator. It offers a thorough evaluation of key parameters, highlighting the advantages of the proposed LDO in terms of energy efficiency and sustainability. In addition, Figure 17 explains the peak voltage and Figure of Merit (FoM) for the proposed LDO regulator integrated with an innovative analog current switch structure. The FoM is a critical index, measuring the performance of the LDO regulator in a balanced view considering both efficiency and sustainability. A comparative analysis, considering the FoM values, provides further evidence of the excellent of the proposed LDO regulator over the conventional ones. The comparison substantiates the fact that proposed LDO regulator not only improves the energy efficiency but also offers advanced ESD protection at the IC level, ensuring high reliability under potential ESD events. Consequently, the proposed LDO regulator's unique design, encompassing an analog current switch structure and an embedded ESD protection circuit, holds potential for advancing sustainable development in the field of power electronics.

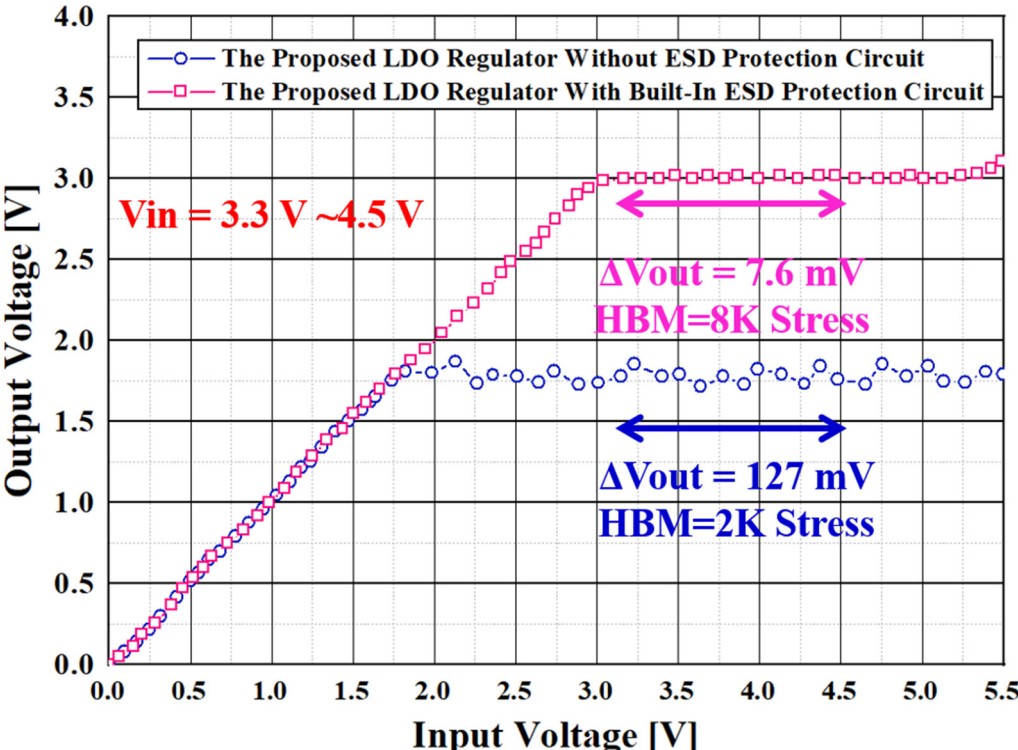

**Figure 13.** Line regulation measurement result of transient current-sensing structure LDO regulator (embedded with/without ESD protection circuit).

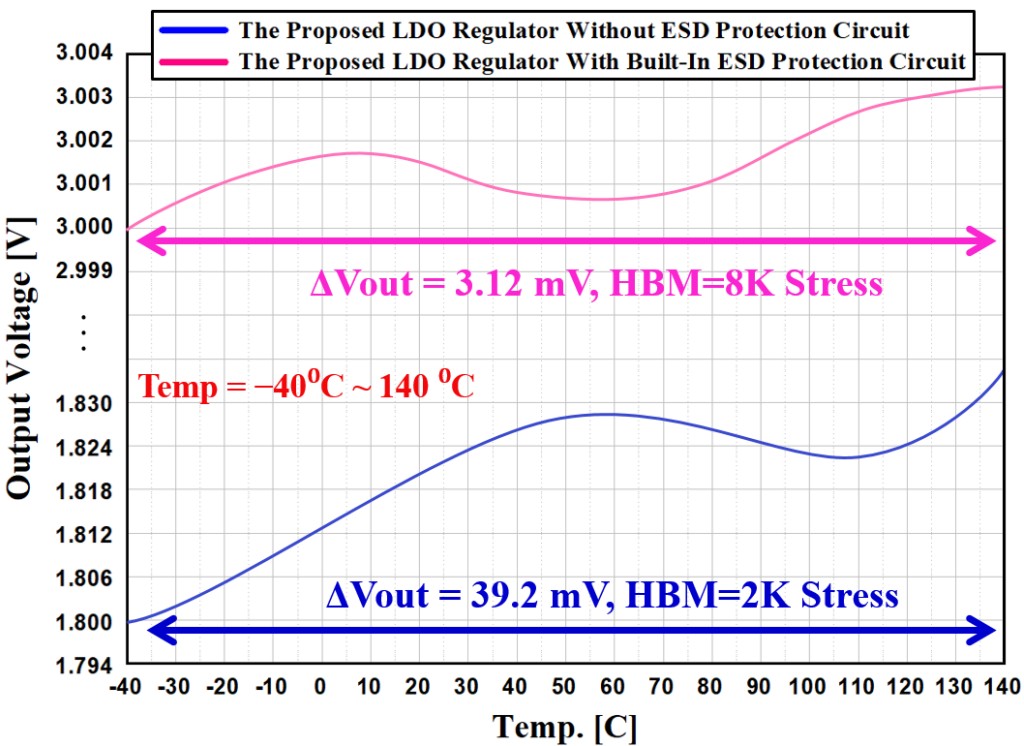

**Figure 14.** Temperature measurement result of transient current-sensing structure LDO regulator (embedded with/without ESD protection circuit).

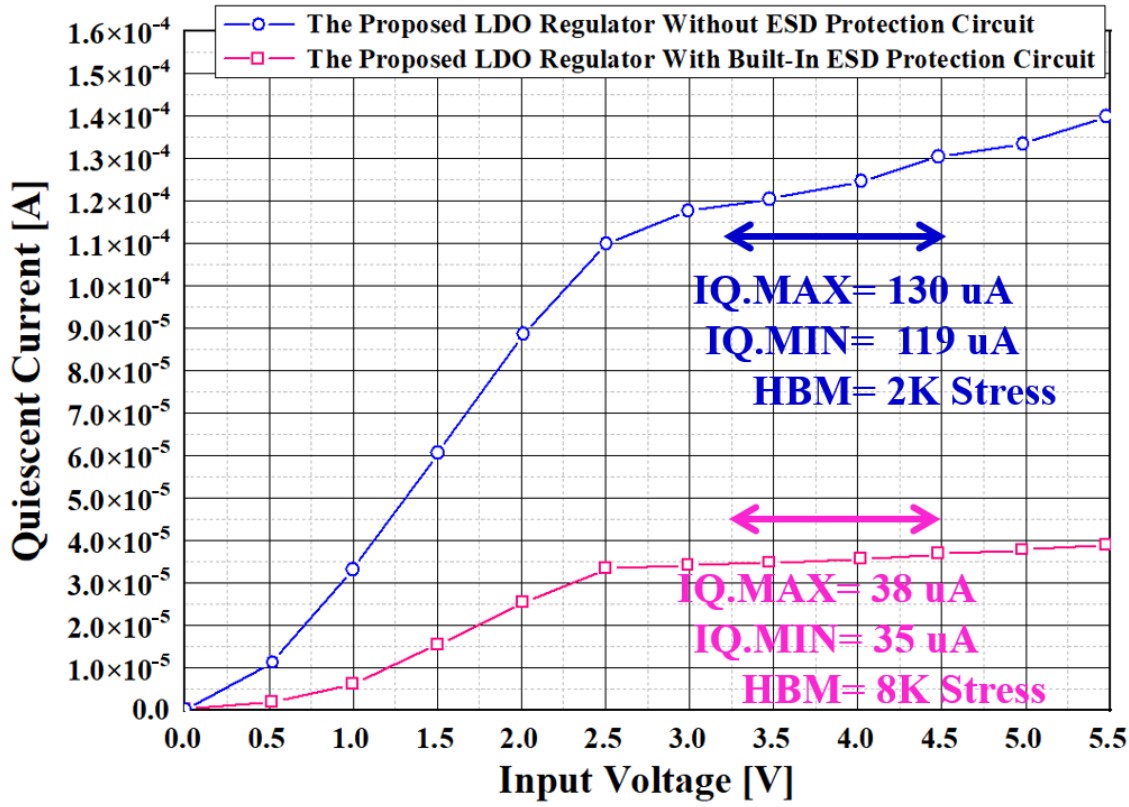

**Figure 15.** Quiescent current result of transient current-sensing structure LDO regulator (embedded with/without ESD protection circuit).

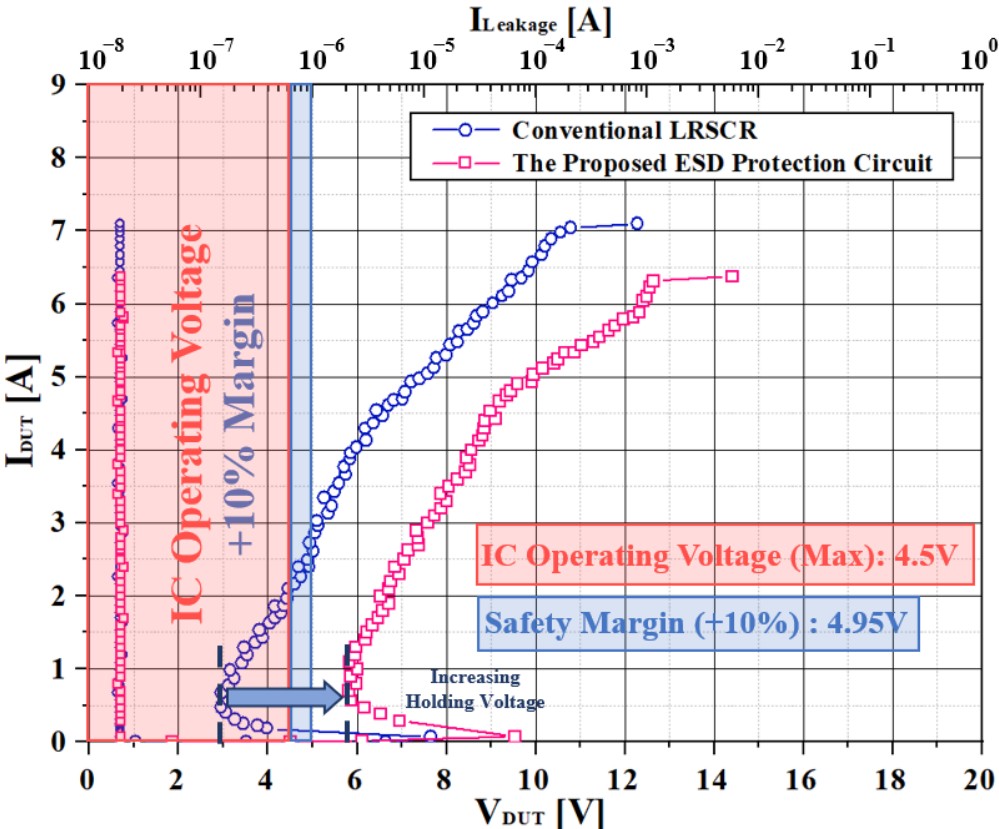

**Figure 16.** TLP I–V characteristics of the proposed ESD protection circuit.

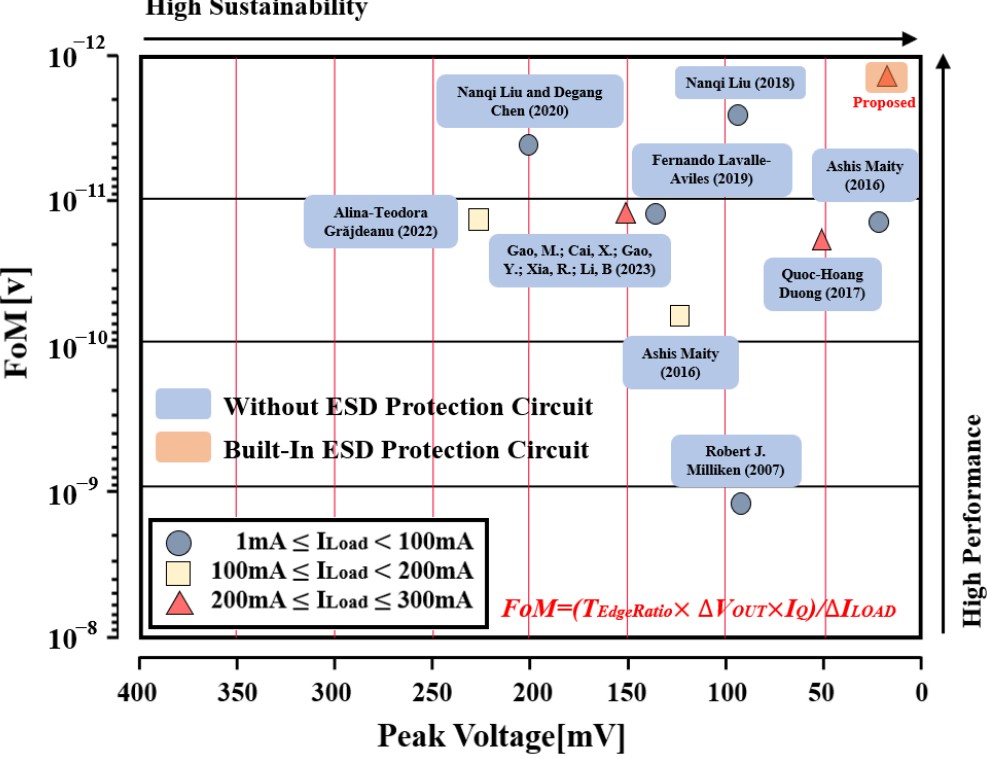

**Figure 17.** Comparative analysis of performance and sustainability between the proposed and conventional LDO regulators.

**Table 1.** Performance comparison of proposed LDO regulators and conventional LDO regulators.

| Measurement | This Work | [1] | [2] | [3] | [4] | [5] | [6] | [7] | [8] | [9] |
|---|---|---|---|---|---|---|---|---|---|---|
| Technology (μm) | 0.13 | 0.65 | 0.18 | 0.13 | 0.13 | 0.18 | 0.35 | 0.13 | 0.13 | 0.18 |
| Supply Voltage(V) | 3.3–4.5 | 1.05–1.2 | 2–5 | 1.2–1.5 | 1.05–2.0 | 1.4 | 3 | 1.2–1.5 | 1.4 | 1.5 |
| Output Voltage (V) | 3 | 0.9 | 1.8 | 1 | 1.0 | 1.2 | 2.8 | 1 | 1.2 | 1.2 |
| Load Current: $I_{MAX}$ (mA) | 300 | 20 | 300 | 50 | 300 | 50 | 50 | 100 | 20 | 100 |
| Quiescent Current (μA) | 38 | 65 | 66.4 | 42 | 14-120 | 0.9-83 | 65 | 6.2 | 100 | 242 |
| Load Transient ($I_{LOAD)}$ Rising) (mV) | 21 | 35 | 86 | 140 | 56 | 18 | 90 | 234 | 90 | 125 |
| Load Transient ($I_{LOAD)}$ Falling) (mV) | 19 | 88 | 67 | 80 | 24 | 7 | 15 | 170 | 95 | 65 |
| Load Regulation (mV) | 6.72 | - | 1.94 | 0.01 | 0.006 | 0.14 | - | - | - | 0.14 |
| Line Regulation (mV) | 7.6 | - | 0.55 | 0.3 | 0.44 | 7.25 | - | - | - | 12.3 |
| COUT (pF) | $5 \times 10^5$ | 0–100 | 5 | 0–400 | $1 \times 10^6$ | $47 \times 10^4$ | 0–100 | $0$–$1 \times 10^6$ | 1.4 | 100 |
| Edge time (ns) | 500 | 5 | 300 | 100 | $1 \times 10^3$ | 400 | $15 \times 10^3$ | 1000 | 10 | 300 |
| FOM (V) | $1.33 \times 10^{-13}$ | $3.25 \times 10^{-12}$ | $1.02 \times 10^{-11}$ | $1.18 \times 10^{-11}$ | $2.24 \times 10^{-11}$ | $1.66 \times 10^{-12}$ | $1.76 \times 10^{-9}$ | $1.45 \times 10^{-10}$ | $4.5 \times 10^{-12}$ | $9.08 \times 10^{-11}$ |
| Year | 2023 | 2020 | 2023 | 2019 | 2017 | 2015 | 2007 | 2021 | 2018 | 2015 |

## 5. Conclusions

In this paper, to test the sustainability of LDO regulators in terms of power semiconductors, it was verified that the sustainability of integrated circuit operation varies greatly depending on the presence or absence of an ESD protection circuit. Another objective of this study was to validate the battery-life extension and electro-static discharge technologies utilized in highly integrated and tiny mobile devices. In terms of sustainable power semiconductors, LDO regulators with built-in ESD protection circuits are a very effective way to protect against ESD situations. Protection against ESD events via diodes cannot protect against large-level ESD events, not just areas. As the current semiconductor process is miniaturized, the development of ESD protection circuits is considered essential compared to diodes. As a result, in this paper, a design using an ESD protection circuit was carried out for the sustainability of power semiconductors. The following is a summary of the findings. First, even with a higher load current, the LDO regulator proposed in this study exhibited a superior transient response and low bias current characteristics. Second, ESD is a very fast (nanosecond) charge transfer phenomenon, and the LDO regulator proposed in this study is proven to meet the ESD requirements of a low-voltage integrated circuit by embedding an SCR-based ESD protection circuit to prevent damage to the IC due to an electrostatic discharge. An LDO regulator must have a stable peak voltage, even when the load current fluctuates. An undershoot voltage of 21 mV and overshoot voltage of 19 mV were observed even at a load current of 300 mA. Consequently, it was proved that the proposed LDO regulator with the transient current-sensing structure efficiently controlled the peak voltage according to the load current. In addition, it was verified that an LDO regulator embedded with an ESD protection circuit stably protected the internal circuit system in ESD situations. The proposed LDO regulator embedded with an ESD protection circuit secured the immune characteristics of HBM = 8 KV. The sustainability of the power semiconductor circuit is always at risk for circuit operation in ESD situations. Consequently, it was confirmed that the LDO regulator has an embedded ESD protection circuit, providing high reliability and stable output voltage even at a large load current. Also, the development of an LDO regulator with an embedded ESD protection circuit was able to verify the sustainability of power semiconductors. In terms of the sustainability of power semiconductors, it will be possible to more effectively design the protection circuit configuration of integrated circuits in the future [37,38].

**Author Contributions:** Conceptualization, S.-W.K.; Methodology, S.-W.K.; Validation, S.-W.K., J.-M.L. and Y.-S.K.; Investigation, S.-W.K. and S.-G.J.; Resources, J.-M.L.; Writing—original draft, S.-W.K.; Writing—review & editing, S.-G.J. and J.-M.L.; Supervision, Y.-S.K.; Project administration, Y.-S.K.; Funding acquisition, Y.-S.K. All authors have read and agreed to the published version of the manuscript.

**Funding:** This research received no external funding.

**Institutional Review Board Statement:** Not applicable.

**Informed Consent Statement:** Not applicable.

**Data Availability Statement:** No new data were created or analyzed in this study. Data sharing is not applicable to this paper.

**Acknowledgments:** This work was supported by Korea Evaluation Institute of Industrial Technology (KEIT) grant funded by the Ministry of Trade, Industry & Energy (20009739, "Development of Low Noise 3phase BLDC Motor Drive SoC for Electric Vehicles with Power Switch and Hall Sensors") and supported by National R&D Program through the National Research Foundation of Korea (NRF) funded by Ministry of Science and ICT (2021M3H2A1038042).

**Conflicts of Interest:** The authors declare no conflict of interest.

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
