# Peer review of "Design of Destruction Protection and Sustainability Low-Dropout Regulator Using an Electrostatic Discharge Protection Circuit"

_sustainability, doi:10.3390/su151310126_

Round 1
Reviewer 1 Report
The manuscript needs to be revised according to the above questions before publication of this manuscript in the Journal.
1) The authors should define the novelty of the problem at the end of the introduction.
2) English of the paper should be polished carefully.
3) Authors are encouraged to discuss the possibility to use Machine learning models based on the following recent works: [(a) “Machine learning models for predicting the compressive strength of concrete containing nano silica”, Computers and Concrete, 30(1), 33-42.; (b) “Predicting elemental stiffness matrix of FG nanoplates using Gaussian Process Regression based surrogate model in framework of layerwise model”, Engineering Analysis with Boundary Elements, 143, 779-795.].
4) Figs. 13 and 14 should be more discussed.
5) Conclusion section is poor. Some applications of the model and future scope should be included.
2) English of the paper should be polished carefully.
Author Response
Dear reviewers and editorial staff in MDPI Sustainability,
We thank the editor and the reviewers for the opportunity to submit a revised version of our paper titled “Design of Destruction Protection and Sustainability for ESD Discharge of LDO Regulator Using ESD Protection Circuit.” Through the accurate comments made by the reviewers, we better understand the critical issues in this paper. We have revised the manuscript according to the Reviewer’s suggestions. We hope that our revised manuscript will be considered and accepted for publication in the MDPI Sustainability. This revision contains additional information that was missing from the previously reviewed paper. We thank you for your insightful review to improve the manuscript. We have carefully considered your comments, and we hope that our revisions are satisfactory.
Sincerely,
Sang-Wook Kwon

Reviewer 2 Report
This paper presents an LDO regulator approach with high reliability against ESD situations using an embedded silicon-controlled rectifier based ESD protection circuit. Although the topic seems interesting, there are several comments and concerns as follows that shall be addressed before publication:
1. Authors should avoid using acronyms in the abstract section or they shall be defined beforehand.
2. Please revise the introduction to remove any lumpsum citations, i.e. [1-9]. Such a citation does not add any value to the paper. It is highly recommended to describe each reference shortly and discuss its contributions or advantage/disadvantages compared to the proposed method.
3. The introduction section is too short. Please elaborate the problem, current methodologies to solve it, the lack of previous investigations and the gap that the proposed scheme tries to cover. What are the merits of the proposed approach in comparison with the previous literature? Currently, the contribution of the paper is unclear.
4. Figure 1 is ambiguous. Where does it come from? Please add a reference for that or if it is created by the authors, it should be properly described. For example, the x-axis label is “Voltage Level”, but for higher voltages, there is no “Reliability” value in case of “Regulators with ESD protection”. Furthermore, the y-axis label should be changed from “High Reliability” to simply “Reliability”
5. What are the limitations of the proposed method. Please clarify.
6. It is recommended to discuss future works in the conclusion section.
Author Response

(The authors gave the same response as above.)

Reviewer 3 Report
Dear Authors,
General comment
The paper presents interesting experimental results. Low-dropout Regulators are usually the optimal choice based on dropout voltage. In my opinion, the paper has several disadvantages. It's hard to judge the performance of the proposed method.
An Equipment
It is not clear what is the using equipment. The detailed type, specification, and manufacturer of involved devices should be provided. The performance parameters such as S/N ratio and repeatability also need to be measured using the proposed technique. I suggest eliminating this disadvantage.
Uncertainty of the Measurement
I see the measurement results, but what is their uncertainty? These results should be verified.
In general, the measurement uncertainty is an important parameter. I suggest eliminating this disadvantage. In order to enable the comparison of these results of measurements, they should be reported (in accordance with the Guide to the Expression of Uncertainty in Measurement*) along with the values of their uncertainty. The Expanded Uncertainty is the internationally accepted basis to assess the quality of measurements.
*) Guide to the Expression of Uncertainty in Measurement. International Standardization Organization (ISO), Geneva 1995.
The Authors ought to estimate these values, which will allow determining the "accuracy" of the proposed method and its practical utility. In the discussion section, I would expect to see an analysis of the possibilities to reduce the measurement uncertainty.
Other
- The conclusion is poor,
- There are a few acronyms that are not defined in the text (e.g. LDO,...).
- The citations in the text are in the wrong order.
Line 43-44 [1-9],
Line 110 [29-33],
Line 165 [15-22],
Line 194 [10-14,
Line 276 [34-37],
Line 291 [23-28],
Line 292 [1-9].
My Conclusion
The paper must be supplemented. Please carefully check all paper. The idea is good, but it needs to be polished.
-
Author Response

(The authors gave the same response as above.)

Reviewer 4 Report
The whole paper is written very nicely and has major significant content. But still, I have a few more questions that need justification.
- Why is the integration of an ESD protection circuit into the semiconductor circuit considered an important aspect of sustainable power semiconductors?
- What is the purpose of the thesis mentioned in the abstract?
- How does embedding an ESD protection circuit at the integrated circuit (IC) level contribute to high reliability in power semiconductors?
- What type of ESD protection circuit is utilized in the proposed LDO regulator?
- What are the advantages of using a silicon-controlled rectifier (SCR)-based ESD protection circuit in the I/O CLAMP and POWER CLAMP?
- How does the LDO regulator effectively control the output voltage based on load current?
- What is the significance of maintaining stable output voltage in the presence of ESD surges?
- What are the specific results obtained from implementing the proposed LDO regulator with an embedded ESD protection circuit in a 0.13μm BCD process?
- How does the proposed LDO regulator perform under a load current of 300 mA in terms of undershoot and overshoot voltage?
na
Author Response

(The authors gave the same response as above.)

Round 2
Reviewer 2 Report
No further comments.